# Barkhausen Noise Emission in Hard-Milled Surfaces

**DOI:** 10.3390/ma12040660

**Published:** 2019-02-22

**Authors:** Miroslav Neslušan, Anna Mičietová, Branislav Hadzima, Branislav Mičieta, Pavel Kejzlar, Jiří Čapek, Juraj Uríček, Filip Pastorek

**Affiliations:** 1University of Žilina, Univerzitná 1, 01026 Žilina, Slovakia; anna.micietova@fstroj.uniza.sk (A.M.); branislav.hadzima@rc.uniza.sk (B.H.); branislav.micieta@fstroj.uniza.sk (B.M.); juraj.uricek@fstroj.uniza.sk (J.U.); filip.pastorek@rc.uniza.sk (F.P.); 2Department of Material Science, Faculty of Mechanical Engineering, Technical University in Liberec, Studentská 1402/2, 46117 Liberec, Czech Republic; pavel.Kejzlar@tul.cz; 3Faculty of Nuclear Sciences and Physical Engineering, ČVUT Praha, Trojanova 13, 12000 Praha, Czech Republic; jiri.Capek@fjfi.cvut.cz

**Keywords:** Barkhausen noise, hard-milling, tool wear

## Abstract

This paper reports on an investigation treating a hard-milled surface as a surface undergoing severe plastic deformation at elevated temperatures. This surface exhibits remarkable magnetic anisotropy (expressed in term of Barkhausen noise). This paper also shows that Barkhausen noise emission in a hard-milled surface is a function of tool wear and the corresponding microstructure transformations initiated in the tool/machined surface interface. The paper discusses the specific character of Barkhausen noise bursts and the unusually high magnitude of Barkhausen noise pulses, especially at a low degree of tool wear. The main causes can be seen in specific structures and the corresponding domain configurations formed during rapid cooling following surface heating. Domains are not randomly but preferentially oriented in the direction of the cutting speed. Barkhausen noise signals (measured in two perpendicular directions such as cutting speed and feed direction) indicate that the mechanism of Bloch wall motion during cyclic magnetization in hard-milled surfaces differs from surfaces produced by grinding cycles or the raw surface after heat treatment.

## 1. Introduction

Magnetic Barkhausen noise (MBN) is a promising non-destructive technique that could be potentially adapted for fast and reliable surface monitoring. An increasing number of studies can now be found in which MBN is investigated as a function stress state [1,2,3,4], dislocation density [5,6], carbides or nitrides size and distribution [7,8], presence of non-ferromagnetic phases [9], etc. However, this technique is most widely employed for monitoring ground surfaces in real industrial applications. The low MBN values for untouched surfaces are in contrast with the high MBN emission due to thermal over-tempering during grinding. MBN is a product of irreversible discontinuous Bloch walls (BWs) motion during cyclic magnetization. BWs interfere with the stress state as well as microstructure features (such as dislocations, carbides, grain boundaries, non-ferromagnetic particles, etc.) that pin the BWs motion. The high MBN magnitude of over-tempered surfaces after grinding is mainly associated with reduced dislocations and carbide density that is thermally initiated by elevated temperatures, which in turn correspond with decreased pinning strength [7,10]. For this reason, MBN can be easily employed for monitoring components after grinding to prevent early microcracks initiation and premature failure of bodies due to surface over-tempering.

Hard machining (mainly turning and milling) can substitute for grinding cycles. Developments in machine tools, as well as in process technology, raised the industrial relevance of hard machining [11]. Hard machining operations are usually employed for machining components with complicated geometries or as roughing cycles before the finish grinding. The mechanism of chip separation during grinding significantly differs from hard machining. For this reason, the surface states produced by these competitive operations are different. The main distinctions are [12]: (i) a much longer time during which higher temperatures penetrate beneath the free surface for grinding (several times greater tool/workpiece contact), (ii) the average stress over the entire contact in grinding is less than in hard milling and (iii) a deeper penetration of compressive stress in hard milling.

Very high heating rates and rapid self-cooling during hard milling generate the specific state of surface integrity expressed in many terms. Very high MBN values and strong magnetic anisotropy can be found on the milled surface despite the restricted structure transformations and high hardness [13,14,15]. Furthermore, tool geometry changes remarkably with increasing cutting time, which in turn remarkably alters the mechanical and thermal load of the machined surface. This aspect of hard machining is of vital importance since the cutting conditions in real industrial applications are usually kept constant while tool wear progressively develops. Hard milling cycles can suffer from the formation of a white layer (WL) and underlying heat affected zone (HAZ). Inserts of high flank wear (VB) produce a relatively thick near-surface WL, as well as the corresponding subsurface (HAZ) [12,16]. However, the thickness of the HAZ and WLs after hard milling is about one order of magnitude lower than that induced by grinding. Hard-turned or milled WLs are denser and more uniform with a severely strained matrix, whereas ground WLs retain their original appearance [12]. Compared to the bulk, the HAZ produces richer Barkhausen noise emissions (due to tensile stresses, reduced dislocation density and the modification of carbides, i.e., their size, density and morphology), whereas WLs induced by grinding cycles in the near-surface region emit poor MBN due to the existence of a higher volume of retained austenite, compressive stresses and very fine grains [17]. 

Very high MBN values and strong magnetic anisotropy can be found after hard milling, despite the low thickness of WLs, as well as the HAZ [13,14,15]. Furthermore, cyclic magnetisation initiates a specific character of BN bursts. These aspects were discussed in previous papers [13,14,15]. However, some specific aspects have not yet been discussed. For this reason, this extensive study presents very complex and deep insights into hard-milled surfaces from the points of view of microstructure, stress alterations and corresponding MBN emission, and the progressively developed VB. The deep understanding of surface state after hard-milling is crucial to suggest appropriate and reliable concepts for monitoring hard-milled components using the MBN technique.

## 2. Materials and Methods

The experimental study was carried out using heat treated (HT) bearing steel 100 Cr6 with 61 ± 1 HRC hardness. HT of the samples was carried out in industrial conditions. Samples 70 × 30 × 25 mm in size were quenched from an 840 °C austenitizing temperature in a 60 °C oil and tempered afterwards at a 140 °C for 2 h. Hard milling was carried out using a 050Q22-12M 262489 milling cutter (Sandvik Coromant, Sandviken, Sweden) of diameter Ø 50 mm with 2 VB inserts of 0.05, 0.2, 0.4, 0.6 or 0.8 mm. Low and very high VB were employed to increase the time during which the undergoing layer was exposed to severe thermo-plastic deformation. A brief illustration of VB, generation of machined surface and chip separation during hard milling is depicted in Figure 1. Hard milling was carried out using a 050Q22-12M 262489 milling cutter of Ø 50 mm diameter with two VB = 0.05 mm inserts. A cutting depth of a_p_ = 0.25 mm, a feed speed of v_f_ = 0.11 m·min^−1^ (the feed direction corresponds with the axial direction on the milled surface) and a cutting speed of v_c_ = 78.5 m·min^−1^ (the cutting speed direction corresponds to the tangential direction on the milled surface) were used (see Figure 2). 

MBN was measured using a RollScan 300 and MicroScan software (Stresstech, Jyväskylä, Finland) (10 V magnetising voltage, 125 Hz magnetising frequency, 10–1000 kHz MBN pulse frequency range, sensor type S1-18-12-01). MBN was obtained by averaging 10 bursts (5 magnetisation cycles) and refers to the rms (effective) value of the signal. In addition to the conventional MBN parameter (rms value of the signal), the peak positions (PP) of the MBN envelopes were analysed. The PP of MBN refers to the position of the magnetic field in which the MBN envelope attains the maximum, which corresponds to the magnetic hardness of the body. MBN was measured in two perpendicular directions corresponding to the process kinematics (see Figure 3). MBN and (XRD) measurements were also carried out on the surface after HT.

To reveal the microstructural transformations induced by hard milling, 10 mm long pieces were prepared for scanning electron microscopy (SEM, UHR Carl Zeiss Ultra Plus, Jena, Germany) and metallographic observations (etched by 5% Nital for 8 s). The microstructure was observed in the direction of cutting speed. 

An X’Pert PRO MPD diffractometer (Panalytical Ltd., Eindhoven, Netherlands) was used to measure lattice deformations in the ferrite phases using the CrKα radiation. The average effective penetration depth of the X-ray radiation was approximately 4 µm. Diffraction angles, 2θ^hkl^*,* were determined from the peaks of the diffraction lines, Kα_1_, of the {211} plane. Diffraction lines, Kα_1_, were fitted to the Pearson VII function and the Rachinger’s method was used for the separation of the Kα_1_ and Kα_2_ diffraction lines. To determine residual stresses (RS), the Winholtz and Cohen method and X-ray elastic constants of ½s_2_ = 5.75 TPa^−1^, s_1_ = −1.25 TPa^−1^ were used. The diffraction patterns were measured using an X’Pert PRO MPD diffractometer in the grazing incidence diffraction geometry (GID) with CoKα radiation. The measurements were performed with a constant effective penetration depth of 1 µm. The Rietveld refinement was used for calculation of the phase composition as well as the mass percentage of retained austenite.

The alternative magneto-optic Kerr effect (MOKE) technique was employed to analyse the specific character of BW motion in the very near-surface layer. MOKE is a specific technique based on the application of a polarised laser beam reflected by the investigated surface in the magnetic field. Its penetration depth is limited to several nanometres. This method can be employed for evaluation of the magnetic properties of nanolayers, the influence of surface reflexivity, anisotropy, the detection of hysteresis loops, appearance, etc.

Three basic configurations can be found depending on magnetic field orientation:longitudinal: magnetisation of a sample is parallel with the laser beam (Figure 4a),transversal: magnetisation of a sample is perpendicular with the laser beam (Figure 4b), andpolar: magnetisation is perpendicular to the sample plane (Figure 4c).

A brief illustration of MOKE equipment is shown in Figure 5 and Figure 6. The measurement was carried out on 20 × 20 × 5 mm specimens in the longitudinal and transversal (perpendicular) directions with a 45° laser beam incidence angle (the difference between the second harmonic components was measured). The other conditions were λ = 670 nm, f = 50 kHz, I = 10 A and 2000 samples per hysteresis loop.

## 3. Results and Discussion

### 3.1. Metallographic and SEM Observations

Hard-milling initiates very high temperatures and superimposes hydrostatic pressure ahead of the cutting edge. There are two theories explaining plastic deformation of hard and brittle steels during turning and milling operations. Thermodynamic theory explains that the formability of the applied mechanical energy is almost exclusively transformed to thermal energy, which heats up the material in front of the cutting edge. Under elevated temperature, the steel softens and obtains a high formability [18,19]. The effect is called self-heating hot machining. Hydrostatic theory is based on the formability as a quantity that is strongly dependent on the stress state. Brittle structure can be plastically deformed under intensive hydrostatic pressure [20]. It is considered that plastic deformation of hardened steel during turning is due to the synergistic effect of both theories. The thermal and superimposed mechanical effects are the decisive factors affecting WL thickness during hard milling. Figure 7 illustrates that the hard-milling cycle produces near-surface WL, which appears white on optical images, and the underlying HAZ, which appears dark on the optical images. However, structural transformations, such as those shown in Figure 7, are driven mainly by the temperature cycle when the near-surface layer is heated above the austenitizing temperature followed by the rapid self-cooling. For this reason, the WL is the region of re-hardened and un-tempered martensite with a hardness exceeding the hardness of untouched bulk [12]. On the other hand, the HAZ represents the region in which temperature does not exceed austenitizing temperature. For this reason, the microstructure of the HAZ is only tempered, which in turn corresponds to a lower hardness compared to the bulk and the WL [12]. It is well known that inserts of low VB produce thin and localized WL and HAZ [16]. The WL and HAZ become continuous and increase in thickness with progressively developed VB. Figure 8 shows that the martensite matrix in the near-surface region was preferentially oriented in the direction of cutting speed. Multiple SEM scans indicate that the thickness of the preferentially oriented layer increased with VB. Figure 9 represents the chemical analysis of the martensite matrix and the embedded particle. It is clearly shown that the particle embedded in the matrix should be referred to as carbides due to the increased intensity of carbon and chromium compared with the martensite matrix. Such a finding also confirms XRD patterns such as those depicted in Figure 10. Figure 8 also shows that some carbides were severely strained in the cutting direction due to superimposing high temperatures and hydrostatic pressure in this region. 

The hard and brittle carbides in the near surface region can behave in a malleable manner when the synergistic effects of elevated temperatures and severe deformation take place. Hosseini [21] and Guo [12] reported that carbides in the WL have an elliptical shape with the long axis aligned with the deformation direction. On both sides of the carbide, the microstructure was reoriented along the axis of the carbide.

Comparing the images in Figure 7 and Figure 8, it was found that the thickness of WL was more than the thickness of the preferentially oriented surface. Preferential orientation of the matrix was due to the elevated temperatures and the superimposing plastic deformation, whereas WL presence was due to the predominating temperature cycle; as a result, temperatures exceeding the austenitizing temperature were followed by the rapid self-cooling. The depth in which severe plastic deformation during machining took place is usually limited whereas elevated temperatures penetrated much deeper due to good thermal conductivity of the bearing steel. 

### 3.2. XRD Measurements

The Rietveld refinement was done for all diffraction patterns (illustrated in Figure 10) with ferrite and martensite phases. The R_p_, R_wp_ and χ^2^ profile residual factors were customarily used to characterize both the full pattern decomposition and Rietveld refinement quality. In all cases, the profile residual factors had smaller values for the refinements with the martensite phase. 

Hard machining cycles usually produce mainly compressive stresses that penetrate quite deep beneath the free surface [22]. Figure 11a confirms that valuable compressive stresses could also be found in the near-surface region in both directions. Furthermore, remarkable stress asymmetry was due to process kinematics when the cutting speed (tangential direction) is one order faster than the feed speed (axial direction). The maximum compressive stresses were found for VB = 0.2 mm in the tangential direction and these stresses decreased with increasing VB. However, the evolution of residual stresses with VB did not exhibit a strong correlation. 

The measured volume of retained austenite after HT (before hard milling) was 19%. The volume of retained austenite in the WL gently dropped down for VB = 0.05 and 0.2 mm, and remarkably increased for VB = 0.4, 0.6 and 0.8 mm. Hosseini [23] explains the decrease in retained austenite volume at lower temperatures due to two synergistic effects: (i) decomposing of the austenite into ferrite and cementite due to generated heat, or (ii) strain-induced martensitic transformation causing austenite to transform into martensite during plastic deformation. Additionally, Ramesh [24] reported that the decrease of retained austenite volume is associated with the predominant effect of plastic deformation. On the other hand, the predominant effect of elevated temperature initiates a remarkable increase in retained austenite volume, as shown in Figure 11b. 

The thickness of the WL and retained austenite volume were a function of VB. Flank wear land represents the path within the machined surface that undergoes the severe plastic deformation at elevated temperatures. Two basic aspects of more developed VB should be discussed: (i) a more developed VB corresponds to a longer time period during which the machined surface undergoes severe plastic deformation at elevated temperatures, and (ii) the temperature in the tool/workpiece interface increases with VB due to increasing normal and shear stresses [25]. Both aspects contributed to the higher volume of retained austenite and thicker WL (as well as HAZ) with increasing VB (Figure 11b). 

### 3.3. MBN Measurements

WL after hard machining (especially turning and milling) is generally described as consisting of a refined structure with a grain size in the order several tens of nanometres [26,27]. For this reason, the high magnitude of MBN pulses (see Figure 12) and the corresponding very high MBN values (in the tangential direction) are very unusual for such a hard body. A heat-treated surface emits MBN at about 100 mV, whereas a hard-milled surface-produced insert of VB = 0.05 mm is 6 times higher despite compressive stresses and a quite thin HAZ. It is worth mentioning that the surface made of hardness 62 HRC bearing steel 100Cr6, heavily over-tempered during grinding, does not emit MBN at more than 350 mV and has a very thick HAZ with remarkable thermal softening [17]. As was previously reported [13,14,15], the main cause can be seen in the specific structure and the corresponding domain configurations formed during rapid cooling following surface heating. Hard milling generates high-temperatures exceeding the Curie temperature in the cutting. It is well known that all ferromagnetic materials show deterioration of magnetism-related properties, such as magnetization and magnetostriction, with increasing temperature as a result of the gradual loss of magnetic order when approaching the Curie temperature, T_C_. A new magnetic order is formed during rapid cooling. Due to remarkable stress anisotropy domains, the corresponding domain walls are not randomly oriented, but rather preferentially oriented in the direction of the cutting speed (tangential direction) at the expense of the perpendicular feed direction (axial direction) (see Figure 8). Strong magnetic anisotropy originates from the corresponding stress anisotropy (see Figure 11a), and superimposes a rapid heating/self-cooling temperature cycle. The constrained time period during which the machined surface undergoes severe plastic deformation at elevated temperatures avoids deeper penetration of structure transformations initiated by hard-milling, especially at a low level of flank wear (VB = 0.05 mm). Therefore, the very high MBN after hard-milling in the tangential direction cannot be associated with either the thermal softening or the stress state. It is well known that compressive stresses tend to decrease MBN. However, such behaviour is valid for uniaxial stress states only. As soon as biaxial or multiaxial stress states take place, the influence of compressive stresses in an analysed direction is compensated for by the influence of stress in the perpendicular direction (or directions) [28]. Krause [28] found that a single easy axis material consists of an isotropic aligned population of moments, giving the background upon which a population of moments with relative orientations is superimposed that results in a net moment within the sample. A dual system consists of a second population of moments with orientations resulting in a net moment with orientation between the individual moments of each population.

The preferential orientation of BWs in the direction of the cutting speed remarkably affects the appearance of MBN bursts due to the specific character of BW motion during cyclic magnetisation. Conventional MBN bursts emitted by the isotropic surface, such as that after HT, are illustrated in Figure 12. As soon as the first weak MBN pulse in the time scale was detected, the amplitude of MBN pulses gradually increased with time and so too the corresponding strength of magnetic field. This amplitude attained a maximum and decreased afterwards. 

Matrix precipitates, grain boundaries and free surfaces usually produce magnetic dipoles [29,30]. Then, the secondary domain structure in the form of closure (reversed) domains can be found at the boundary of precipitates [31] (or other non-ferromagnetic particles) and the matrix, on the free surface [29,30] and grain boundaries (to reduce or diminish the magnetostatic energy). Closure domains decrease in size with increasing magnetic field; however, some of them could retain the domain structure despite very high magnetic fields being employed [31]. As soon as the magnetic field is reversed, subsequent domain growth occurs in the opposite direction and BW motion is preferentially initiated on the already presented closure domains. 

Conventional MBN bursts, such as those illustrated in Figure 12, occur in all cases when the magnetic field for domain nucleation, H_n_, is less than the magnetic field needed for their growth, H_g_. Alternatively, the very high MBN emission for the hard-milled surface is due to H_n_ > H_g_ [29]. The high H_n_ is due to the strong surface texture after hard-milling; a small misorientation of neighbouring grains decreases the density of magnetic dipoles. The domains in the near-surface region stay aligned parallel with the machined surface. Cyclic magnetisation only changes their alignment to the opposite direction. MBN in this layer occurs in the form of a single massive MBN event (or a few MBN jumps). The specific mechanism of BW motion is exhibited in Figure 12 (for the tangential direction), in which an abrupt and immediate increase in MBN magnitude could be found during magnetisation.

Figure 12 also shows that the mechanism of magnetisation in the axial direction differed from the tangential direction. The axial direction represents the hard axis of magnetisation and the mechanism of BW motion consisted of BW rotation in the initial phase followed by irreversible discontinuous jumps. For this reason, the magnitude of MBN pulses in the axial direction were much lower than that in the direction of the easy axis of magnetisation, which in turn resulted in remarkably lower MBN values, especially at lower VB (see Figure 13). 

Figure 8, Figure 11b and Figure 13 also show that VB and the corresponding alteration in the near surface region (considering the preferential orientation as well as the mass percentage of retained austenite) take a significant role. One might expect that MBN in the tangential direction would increase along with increasing VB since the preferential orientation of the martensite matrix penetrates deeper. However, this aspect was strongly compensated by the increasing retained austenite volume (see Figure 11b). On one hand, increasing retained austenite volume decreased the ferromagnetic martensite volume as the phase contributing to the MBN signal. On the other hand, retained austenite as a non-ferromagnetic phase strongly pinned BW motion in the ferromagnetic phase. Comparing Figure 11b and Figure 13, it can be seen that MBN in the tangential direction was inversely proportional to the volume of retained austenite.

The appearance of MBN envelopes (see Figure 14) strongly corresponds with the MBN values indicated in Figure 13. The MBN envelope peak maximum decreased with more developed VB and the peak position was gently shifted towards higher magnetic fields. Compared to the tangential direction, envelopes for the axial direction were shifted to the higher magnetic field and exhibited a lower magnitude due to the specific preferential domain alignment perpendicular to the axial direction. Moreover, comparing the different directions, remarkable distinctions could be found in the appearance of the ascending and the descending part of the MBN envelops.

MBN envelopes for the tangential direction (Figure 14a) and low VB exhibited a steep increase corresponding with the short rise time of MBN pulses (see also Figure 12a). As soon as the volume of retained austenite increased with VB and the peak height of the MBN envelope progressively dropped, the mechanism of magnetization altered, which in turn made the increase of MBN envelopes moderate for the higher VB. The moderate MBN envelope increase was due to the formation of closure domains near retained austenite. For this reason, the nucleation of closure domains and their subsequent growth took place during cycling magnetisation. 

Figure 15 shows that the PP of MBN (for both directions) envelopes increased gently with VB but was saturated for the higher VB. Furthermore, the PP of MBN envelopes for the tangential direction were significantly higher than those for the axial direction or bulk (see Figure 14 and Figure 15). Higher PP in the tangential direction (compared to bulk; PP for bulk is 0.33 ± 0.08 kA·m^−1^) was due to the higher mechanical strength, and the corresponding magnetic strength, of WL. However, the axial direction was the hard axis of magnetisation; therefore BW (in the preferentially oriented matrix) rotated in the initial phase of magnetisation, followed by its irreversible and discontinuous motion. For this reason, the PP of MBN envelopes in the axial direction was found at stronger magnetising fields than those for the bulk, as well as the tangential direction (see Figure 14 and Figure 15). 

To gain deeper insight into the magnetisation process, especially the nucleation process associated with 90° BWs and the massive MBN pulses originating from 180° BWs in the preferentially-oriented near-surface WL, further analyses were carried out. Martines-Ortiz et al. [32] reported that MBN events near the main peak in the MBN envelope are associated with pure 180° BW motion. They show that the width of this region (dH_180_) is approximately 25% of the entire magnetic field (dH_e_) in which any MBN events occur and name this region R^180^ [32]. The width and position of this region are defined as the MBN envelope maximum ±12.5% of dH_e_. The region (dH_90_) between the first point in which the MBN envelope grows above the background noise to the R^180^ region is attributed to the nucleation and motion of mainly reversed (90° BWs) domains. This region is named R^90^ hereafter. The MBN energy in a certain region (defined by the width *dH*) can be calculated using Equation (1):(1)$$E=\underset{events}{\sum }\int^{\;}mV^{2}\;dH$$

It should be noted that the exact evaluation of the different regions could be debatable. In particular, the boundary between dH_90_/dH_180_ should be considered as a region in which mixed MBN motion could be expected (motion of both 180° and 90° BWs), as reported in a previous study [33]. A simplified model of dH_90_ and dH_180_ distribution on the MBN envelope, such as that reported in Reference [32], provides deeper insight into MBN.

As opposed to the axial direction or HT surface, Figure 16a shows a remarkably reduced R^90^ and dominating R^180^ region (and the corresponding MBN energy; see Figure 17a) for the tangential direction (especially for lower VB) due to the specific mechanism of BW motion. On the other hand, the large area of the R^90^ region and the corresponding E_R^90^ energy for the axial direction (see Figure 16b and Figure 17b) should be associated with the (i) nucleation and subsequent growth of closure domains (motion of 90° BWs originating from deeper layers lying outside of WL), (ii) process of 180° BWs rotation in the WL and (iii) nucleation and subsequent growth of closure domains originating from the presence of carbides (motion of 90° BWs) in the WL (see Figure 8). 

Figure 17a shows that the nucleation of closure domains and their subsequent growth in the tangential direction can still be detected despite the preferential orientation of the WL. Manh [34] reported that the number and the size of MBN events in this magnetisation stage is associated mainly with the presence of magnetic free poles related to the misorientation between the grains and, therefore, to the misorientation angle distribution of the surface. However, due to the preferential orientation of the matrix in the near-surface region, the main source of closure domains (in this particular case) and the corresponding 90° BWs motion can be viewed mainly in: (i) the presence of carbides as preferential sites for closure domains in the very thin WL (see Figure 8), and (ii) the contribution of HAZ or untouched deeper layers with the MBN skin-depth. 

Figure 17b and Figure 18 indicate that the contribution of 180° BWs’ rotation, as well as 90° irreversible motion, are the main sources of R_E^90^ energy to the overall MBN emission since only minor evolution of MBN values shown in Figure 13 strongly correlates with the E_R^180^ energy (see Figure 17b). For this reason, the massive MBN pulses originating from the preferential near-surface matrix dominated and took a major role in MBN emission (especially for the tangential direction). 

It is worth mentioning that the hard-milled surface should be considered a gradient structure. The magnetic hardness or/and preferential orientation corresponds to the microstructure and stress gradient. As Figure 8 illustrates, the near-surface region was nearly parallel with the cutting direction. However, the matrix became tilted with the increasing angle against the free surface with increasing depth beneath the free surface. This means that the contribution of the different layers beneath the surface would be altered.

### 3.4. MOKE Technique Observations

It is shown in Figure 19a,c that the hysteresis curve measured in the tangential direction for VB = 0.05 mm was narrow and nearly angular, while the hysteresis curve measured in the axial direction, was extremely narrowed to zero coercivity and remanence. This behaviour may be explained by the presence of a strong magnetoelastic uniaxial easy axis in the tangential direction. This hysteresis loop appearance corresponds with domain wall dynamics for VB = 0.05 mm when MBN in the thin near-surface layer (in the tangential direction) occurred in the form of a single massive MBN event or a few MBN jumps.

As soon as the preferential orientation of the matrix in the near-surface region was disturbed by increasing retained austenite volume, such as that for VB = 0.4 mm when the matrix contains 30% volume, the typical S–shape of the hysteresis loop was obtained for the tangential retained austenite, as well as the axial directions due to presence of closure domains surrounded by 90° domain walls. The magnetisation process, in this case, occurred over a wider range of magnetic field. Moreover, the saturation value of the hysteresis loops for VB = 0.4 mm was lower than that found for VB = 0.05 mm due to the presence of a higher volume of non-ferromagnetic austenite in the surface.

Information about the magnetisation process in the form of hysteresis loops measured in the longitudinal and transversal direction can be employed for modelling domain wall motion in the form of polar graphs such as those shown in Figure 19b,d and Figure 20b,d. Figure 19b and Figure 20b demonstrate that a small change altering the magnetic field, such as its direction as indicated by blue arrows, initiated a very early abrupt change of magnetisation as a result of an irreversible domain wall jump in the case of tangential direction as the easy axis of magnetisation. On the other hand, Figure 19d, and especially Figure 20d, illustrate a remarkable phase of domain wall rotation followed by irreversible and discontinuous motion in the case of the axial direction as the hard axis of magnetisation. In other words, it is obvious that uniaxial anisotropy promoted domain nucleation and magnetic reversal in one step if the magnetic field was applied in the tangential direction and it supported the coherent rotation of magnetization in the plane, combined with two-step switching if the magnetic field was applied in the axial direction [35]. Such findings fully correspond with the character of MBN emission and especially the shape of MBN envelopes (see Figure 14).

## 4. Conclusions

The acceptable surface state expressed in MBN threshold depends on the functionality of components being hard-milled. The possible concept in which a hard-milled surface would be monitored is driven by the relationship between MBN and thickness of the near-surface region altered by the cutting process. As opposed to grinding, the high MBN values and degree of magnetic anisotropy should be linked with the low thickness of altered layers, whereas a progressive decrease of MBN should be linked with the thicker region of the near-surface containing increasing volume of retained austenite and undergoing severe plastic deformation at elevated temperatures. Due to the preferential orientation of the near surface region (tangential direction) and the corresponding BWs alignment MBN in this layer occurs in the form of a single massive MBN event. The remarkably lower MBN for the axial direction is associated with the nucleation and subsequent growth of 90° BWs, as well as the process of 180° BWs rotation in the WL. The hysteresis loops, as well as the polar graphs, prove that a small change altering the magnetic field initiates a very early abrupt change of magnetisation as a result of irreversible domain wall jump in the tangential direction, whereas the coherent BWs rotation combined with two-step switching can be found for the axial direction. 

## Figures and Tables

**Figure 1 materials-12-00660-f001:**
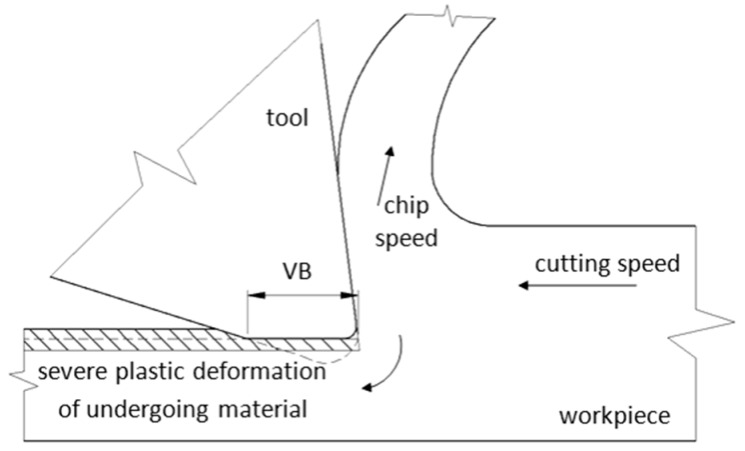
Plastic deformation of undergoing layer and chip separation during hard milling.

**Figure 2 materials-12-00660-f002:**
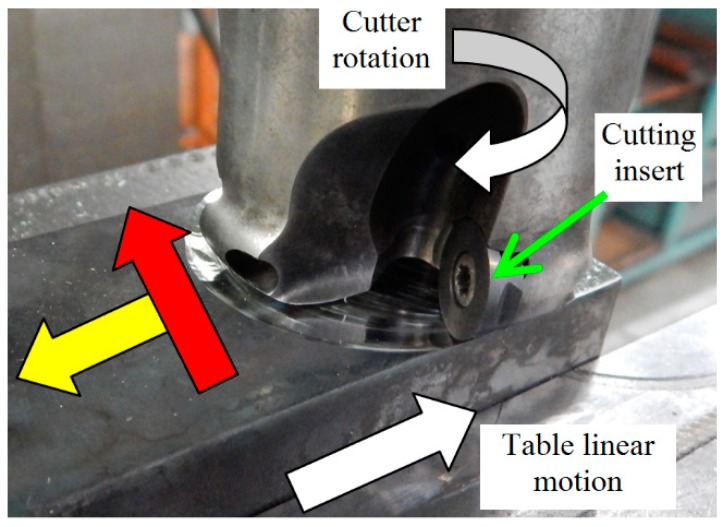
Illustration of hard milling process kinematics, (red arrow—direction of v_c_, tangential; yellow arrow—direction of v_f_, axial).

**Figure 3 materials-12-00660-f003:**
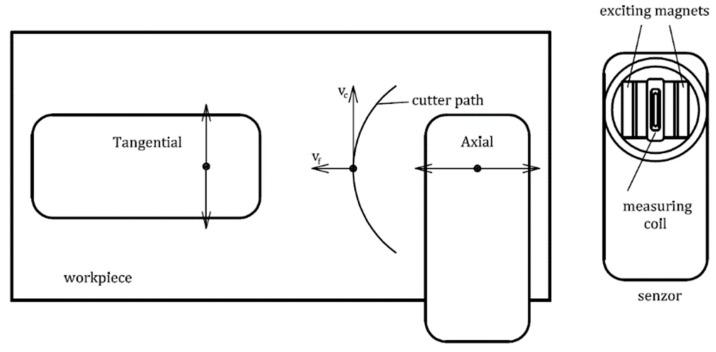
MBN sensor positioning.

**Figure 4 materials-12-00660-f004:**
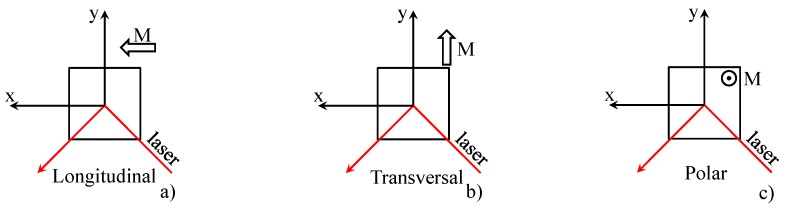
3 basic configurations during measurement of Kerr rotation. (**a**) Longitudinal direction; (**b**) transversal direction; (**c**) polar direction.

**Figure 5 materials-12-00660-f005:**
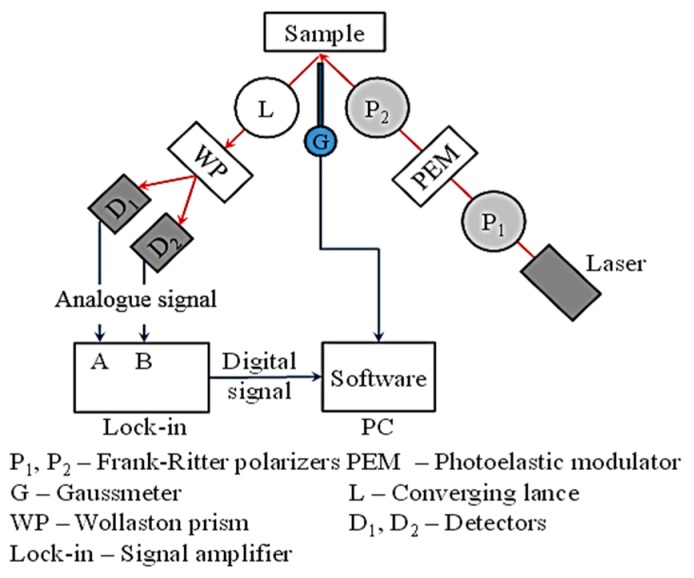
Brief illustration of MOKE.

**Figure 6 materials-12-00660-f006:**
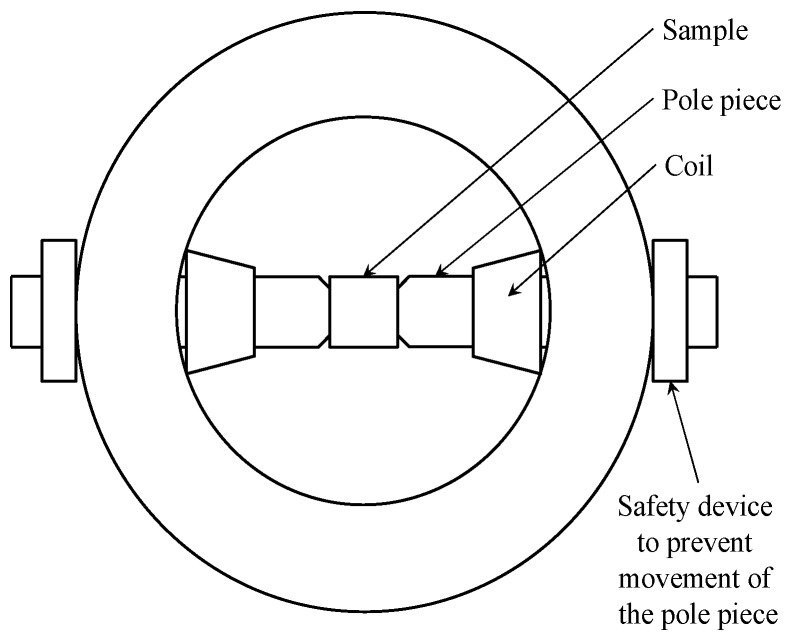
Position of a sample between poles.

**Figure 7 materials-12-00660-f007:**
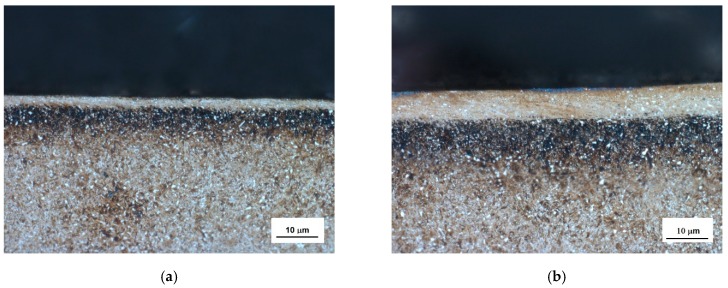
Optical images of milled surface: (**a**) VB = 0.6 mm, and (**b**) VB = 0.8 mm.

**Figure 8 materials-12-00660-f008:**
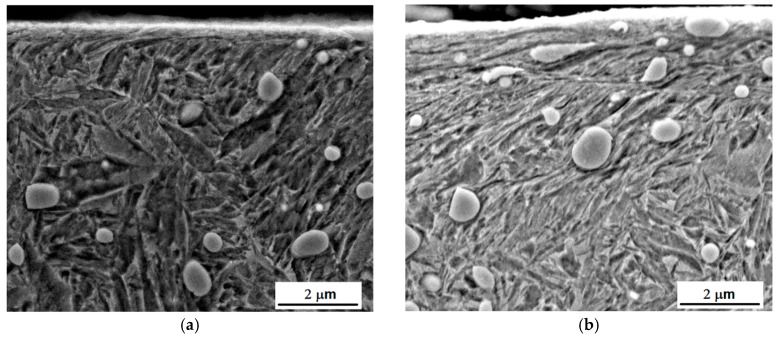
SEM micrographs of milled surfaces: (**a**) VB = 0.05 mm, (**b**) VB = 0.2 mm, (**c**) VB = 0.6 mm, and (**d**) VB = 0.8 mm.

**Figure 9 materials-12-00660-f009:**
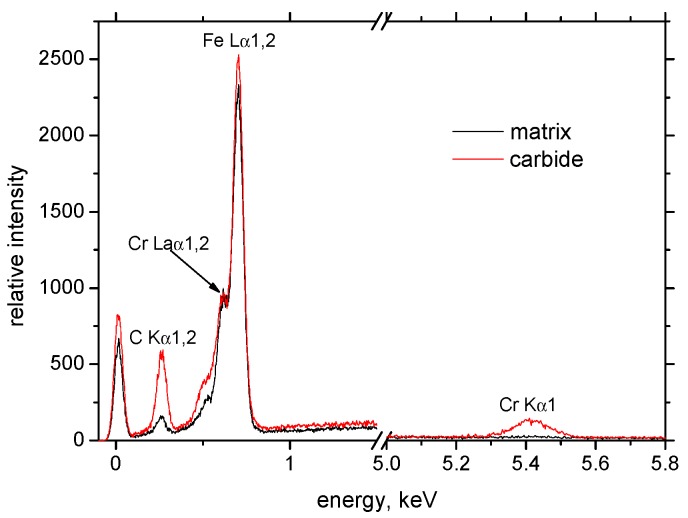
EDS analysis of matrix and carbide.

**Figure 10 materials-12-00660-f010:**
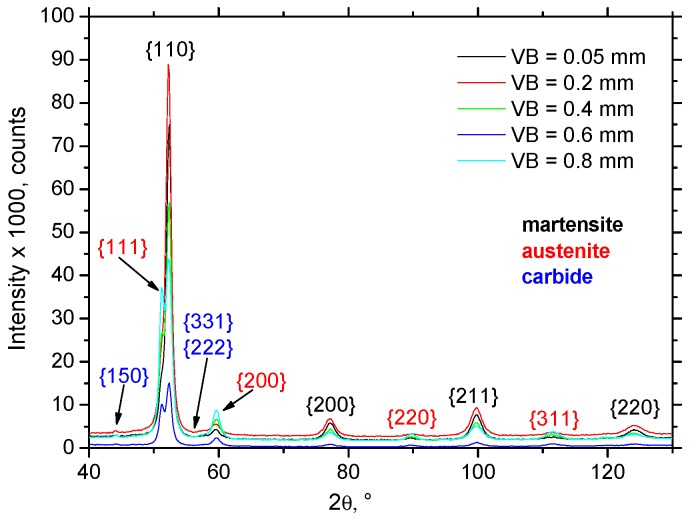
Diffraction patterns from surfaces.

**Figure 11 materials-12-00660-f011:**
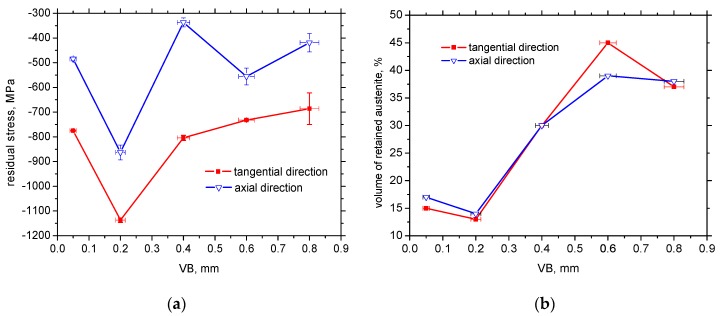
Evolution of residual stresses and retained austenite volume with VB (volume of retained austenite after HT 19%): (**a**) evolution of residual stresses, and (**b**) evolution of retained austenite volume.

**Figure 12 materials-12-00660-f012:**
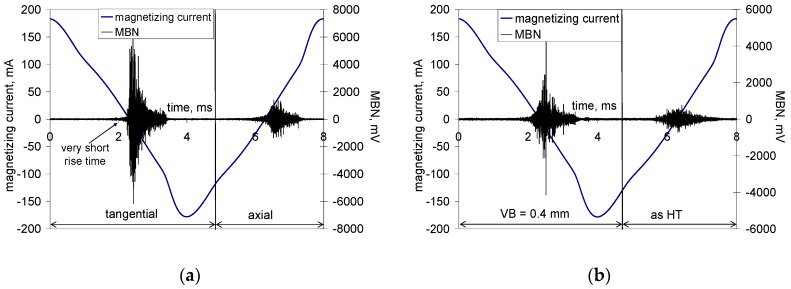
MBN signals: (**a**) for the tangential and axial directions—VB = 0.05 mm, and (**b**) for the tangential direction—VB = 0.4 mm and as HT.

**Figure 13 materials-12-00660-f013:**
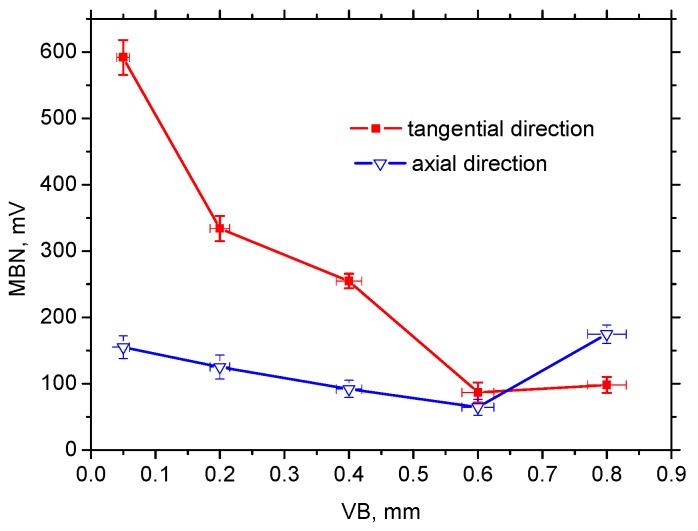
MBN for tangential and axial directions, MBN for HT surface 100 mV.

**Figure 14 materials-12-00660-f014:**
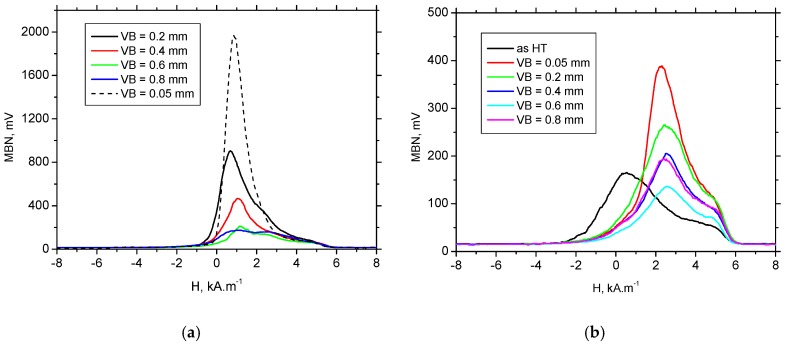
MBN envelopes: (**a**) for tangential direction, and (**b**) for axial direction.

**Figure 15 materials-12-00660-f015:**
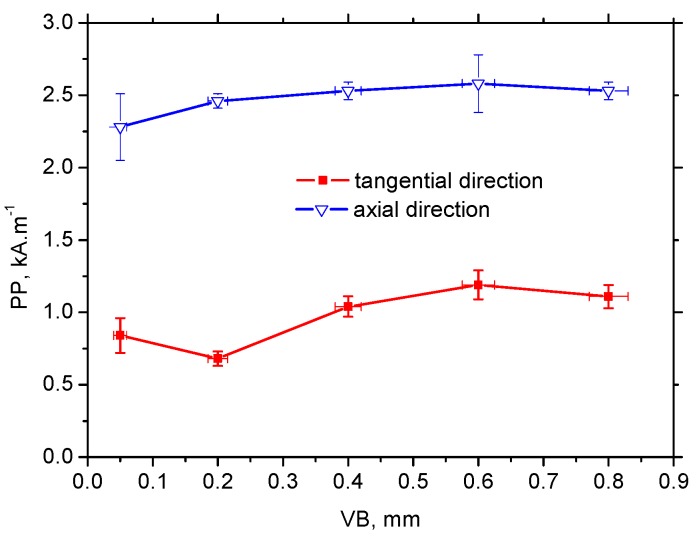
PP of MBN versus VB, PP of MBN for HT surface 0.33 ± 0.08 kA·m^−1^.

**Figure 16 materials-12-00660-f016:**
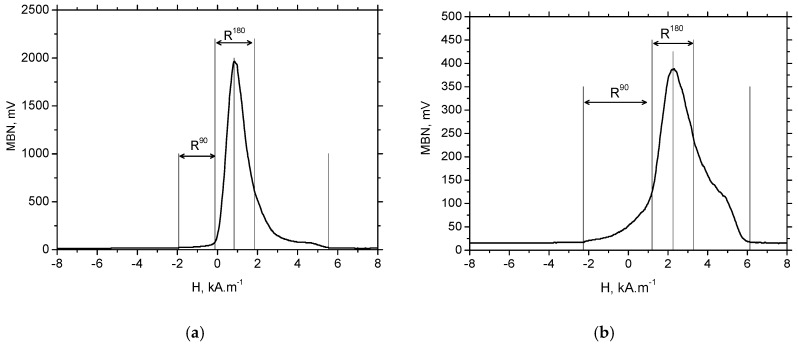
MBN envelopes for VB = 0.05 mm: (**a**) tangential direction, and (**b**) the axial direction.

**Figure 17 materials-12-00660-f017:**
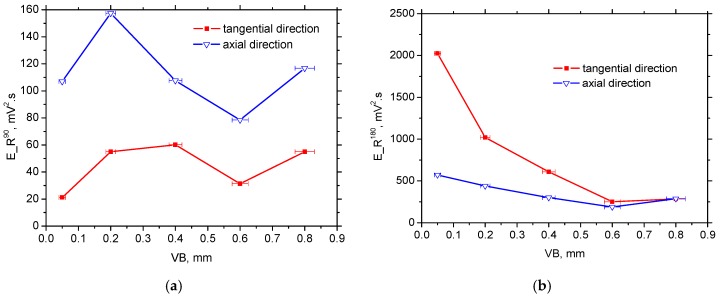
E_R^90^ and E_R^180^ along with VB: (**a**) E_R^90^—60.3 mV^2^·s for HT, and (**b**) E_R^180^—272 mV^2^·s for HT.

**Figure 18 materials-12-00660-f018:**
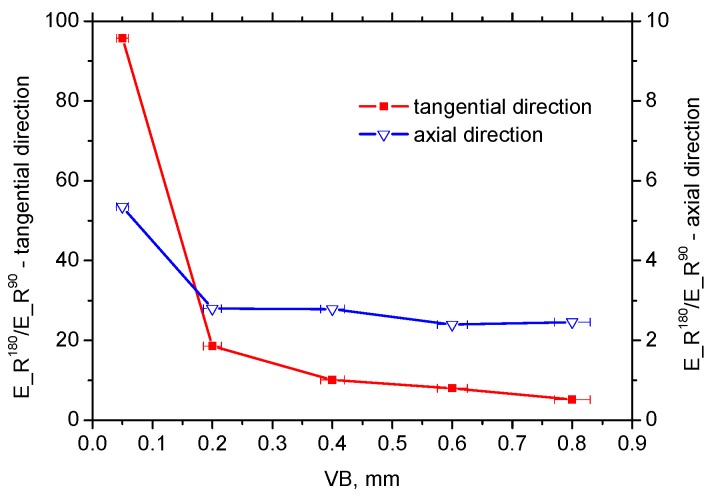
E_R^180^/E_R^90^ along with VB, 4.5 for HT.

**Figure 19 materials-12-00660-f019:**
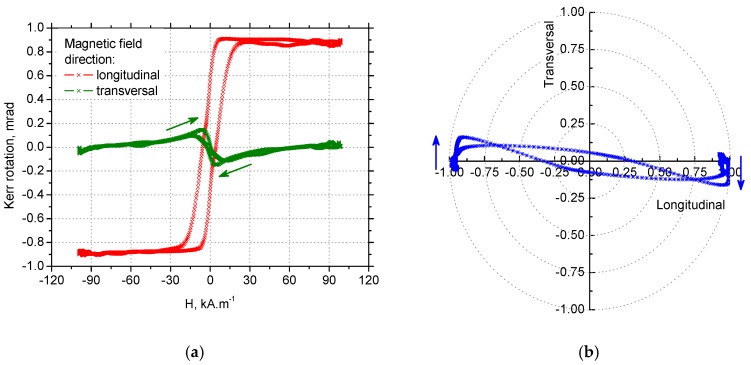
Hysteresis loops and polar graphs for VB = 0.05 mm, normalized: (**a**) hysteresis loops for the tangential direction, (**b**) polar graph for the tangential direction, (**c**) hysteresis loops for the axial direction, and (**d**) polar graph for the axial direction.

**Figure 20 materials-12-00660-f020:**
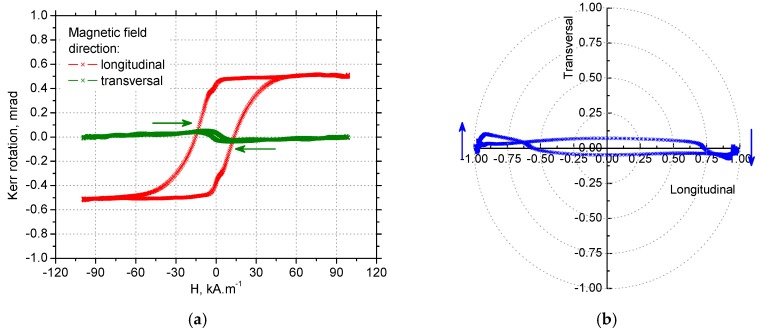
Hysteresis loops and polar graphs for VB = 0.4 mm, normalized: (**a**) hysteresis loops for the tangential direction, (**b**) polar graph for the tangential direction, (**c**) hysteresis loops for the axial direction, and (**d**) polar graph for the axial direction.

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
