# Peer review of "Barkhausen Noise Emission in Hard-Milled Surfaces"

_materials, 2019, doi:10.3390/ma12040660_

Round 1
Reviewer 1 Report
Review of the paper "Barkhausen noise emission in hard-milled surfaces" by Miroslav Neslušan*, Anna Mičietová, Branislav Hadzima, Branislav Mičieta, Pavel Kejzlar, Jiří Čapek, Juraj Uríček, and Filip Pastorek
The research was great but the author do not put to much effort to present it well to audience.
Some couples of observation are here. Apart of them it require a lot of improvements.
Figure 1 point out "plastic deformation of undergoing" how do we know this is plastic deformation ?
Microstructure is very poor monitored, because is almost impossible to suggest "martensite matrix" by OM. Therefore, the Figure 7 and their info needs update
May is is right what they see in Figure 8 "hard and brittle carbides" but tehy should validated by EDS otherwise is not valid info.
How was measured this "The measured volume of retained austenite after HT (before hard milling) is 19%."? because I don not agree the OM measurements.
This part "Matrix precipitates, grain boundaries...already presented closure domains." is not supported by any evidence literature or authors measurements.
How was calibarted "Barkhausen noise " for example sin Figure 10 ?
Again "Figure 8 and Figure 11 also show that VB and the corresponding alteration in the microstructure" they stated about microstructure but not clear evidence.
The conclusion doesn't express clear feedback for this research.
Therefore, I suggest major (hard) revisions. Apart of the above comments the paper should be revised very very carefully and improved accordingly otherwise I will recommend rejection.
Author Response
Response to the reviewers
All changes made in the manuscript (additional texts and corrections) are highlighted yellow color (valid for the manuscript as well as this document).
Reviewer n. 2:
The article describes the characterization of a hard-milling process by means of X-ray diffraction, Barkhausen noise measurements and observations based on the KERR-effect. The motivation of the study is to characterize the hard-milling process and the material parameter relating thereto by the means of Barkhausen noise measurements and to separate the influence by residual stress-, remaining austenite measurements and description of the magnetization behavior.
The study has a good scientific approach, with comprehensive characterization methods. The experimental procedure and the results are presented in a comprehensible manner.
Reviewer: To improve the article the conclusion may be more extensive.
Response: We agree and therefore we added edited more extensive conclusions.
Manuscript: The acceptable surface state expressed in MBN threshold depends on the functionality of components being hard milled. The possible concept in which a hard-milled surface would be monitored is driven by the relationship between MBN and thickness of the near-surface region altered by the cutting process. As opposed to grinding, the high MBN values and degree of magnetic anisotropy should be linked with the low thickness of altered layers, whereas a progressive decrease of MBN should be linked with the thicker region of the near-surface containing increasing volume of retained austenite and undergoing severe plastic deformation at elevated temperatures. Due to the preferential orientation of the near surface region (tangential direction) and the corresponding BWs alignment MBN in this layer occurs in the form of a single massive MBN event. The remarkably lower MBN for the axial direction is associated with the nucleation and subsequent growth of 90° BWs as well as the process of 180° BWs rotation in the WL. The hysteresis loops as well as the polar graphs prove that a small change altering the magnetic field initiates a very early abrupt change of magnetisation as a result of irreversible domain wall jump in the tangential direction whereas the coherent BWs rotation combined with two-step switching can be found for the axial direction.
Reviewer: Line 95: “2” should be written as “two”
Response: corrected
Manuscript: MBN was measured in two perpendicular directions …..
Reviewer: Line 105: RS is not introduced.
Response: corrected
Manuscript: To determine residual stresses (RS), the Winholtz and Cohen ...
Reviewer: Figure 7: Why is the color of the mounting resin different?
Response: We updated images in Fig. 7. However it should be considered that hard machining process produces quite low WL (or HAZ) for low VB (usually difficult to observe in OM – thin and discontinuous). Therefore for better illustration we decided to provide OM mainly for samples milled by the inserts of high VB = 0.6 and 0.8 mm (thicker WL and HAZ) and details in the form of SEM images in Fig. 8. We tried to find the position on the sample surface in which the thickness of WL attains the maximum for better illustration.
MBN emission after hard milling is mainly a function of the thin near surface layer severely strained in the direction of cutting speed (tangential direction). For this reason, we provide details in the form of SEM images about extent of remarkable preferential orientation of the machined surface. Finally it can be found that the thickness of WLs on OM is more than the thickness of preferentially oriented surface. Preferential orientation of the matrix is due to elevated temperatures and superimposing plastic deformation whereas WL is due to predominating temperature cycle as a result temperatures exceeding austenitizing temperature followed by rapid cooling. Influence of severe plastic deformation in machining is limited whereas elevated temperatures penetrates much deeper due to good thermal conductivity of the bearing steel.
Manuscript: Please check the appearance of optical images in Fig. 7 and Fig. 8.
Reviewer: Figure 8: Different magnifications make it difficult to compare the micrographs
Response: Yes, we agree – the magnifications are really different. Therefore we updated images in Fig. 8.
Manuscript: Please check the appearance of images in Fig. 8.
Reviewer: Contaminations can be seen in both images
Response: Yes, we agree – corrected. We updated images in Fig. 8.
Manuscript: Please check the appearance of images in Fig. 8.
Reviewer: Recommend further publications for integration in “References”:
Baak, N.; Schaldach, F.; Nickel, J.; Biermann, D.; Walther, F. Barkhausen Noise Assessment of the Surface Conditions Due to Deep Hole Drilling and Their Influence on the Fatigue Behaviour of AISI 4140. Met. 2018, 8, 720.
Santa-aho, S.; Vippola, M.; Sorsa, A.; Leiviskä, K.; Lindgren, M.; Lepistö, T.:
Utilization of Barkhausen noise magnetizing sweeps for case-depth detection from hardened steel. NDT&E Int. 2012, 52, 95–102.
Response: we integrated the aforementioned publication in the manuscript main body as well as “references”
Manuscript: please check list of references (references 3 and 4) as well as manuscript main body.
… MBN is investigated as a function stress state [1 ÷ 4], dislocation…
Reviewer n. 1:
Review of the paper "Barkhausen noise emission in hard-milled surfaces" by Miroslav Neslušan*, Anna Mičietová, Branislav Hadzima, Branislav Mičieta, Pavel Kejzlar, Jiří Čapek, Juraj Uríček, and Filip Pastorek
The research was great but the author do not put to much effort to present it well to audience.
Some couples of observation are here. Apart of them it require a lot of improvements.
Reviewer: Figure 1 point out "plastic deformation of undergoing" how do we know this is plastic deformation?
Response: Plastic deformation explains thermodynamic theory as well as superimposing hydrostatic theory. Therefore we added the text in which we explain why hard and brittle martensite behaves in malleable manner. Moreover, plastic deformation process confirms also SEM images in Fig. 8 in which remarkable deformation of the near surface martensite matrix as well as carbides can be found.
Manuscript: There are two theories explaining plastic deformation of hard and brittle steels during turning and milling operations. Thermodynamic theory explains that the formability of the applied mechanical energy is almost exclusively transformed to thermal energy which heats up the material in front of the cutting edge. Under elevated temperature, the steel softens and obtains a high formability [18, 19]. The effect is called self-heating hot machining. Hydrostatic theory is based on the formability as a quantity that is strongly dependent on the stress state. Brittle structure can be plastically deformed under intensive hydrostatic pressure [20]. It is considered that plastic deformation of hardened steel during turning is due to the synergistic effect of both theories. The thermal and superimposed mechanical effects are the decisive factors affecting WL thickness during hard milling.
Reviewer: Microstructure is very poor monitored, because is almost impossible to suggest "martensite matrix" by OM. Therefore, the Figure 7 and their info needs update
Response: We updated images in Fig. 7 as well as in Fig. 8. We added Fig. 9 and Fig. 10. However it should be considered that hard machining process produces quite low WL (or HAZ) for low VB (usually difficult to observe in OM – thin and discontinuous). Therefore for better illustration we decided to provide OM mainly for samples milled by the inserts of high VB = 0.6 and 0.8 mm (thicker WL and HAZ) and details in the form of SEM images in Fig. 8. We tried to find the position on the sample surface in which the thickness of WL attains the maximum for better illustration.
MBN emission after hard milling is mainly a function of the thin near surface layer severely strained in the direction of cutting speed (tangential direction). For this reason, we provide details in the form of SEM images about extent of remarkable preferential orientation of the machined surface. Finally it can be found that the thickness of WLs on OM is more than the thickness of preferentially oriented surface. Preferential orientation of the matrix is due to elevated temperatures and superimposing plastic deformation whereas WL is due to predominating temperature cycle as a result temperatures exceeding austenitizing temperature followed by rapid cooling. Influence of severe plastic deformation in machining is limited whereas elevated temperatures penetrates much deeper due to good thermal conductivity of the bearing steel.
Manuscript: Please check the appearance of optical images in Fig. 7 and Fig. 8. Added text: Comparing the images in Figure 7 and Figure 8 it can be found that the thickness of WL is more than the thickness of preferentially oriented surface. Preferential orientation of the matrix is due to the elevated temperatures and the superimposing plastic deformation whereas WL presence is due to the predominating temperature cycle as a result temperatures exceeding austenitizing temperature followed by the rapid self-cooling. The depth in which severe plastic deformation during machining takes place is usually limited whereas elevated temperatures penetrates much deeper due to good thermal conductivity of the bearing steel.
The Rietveld refinement was done for all diffraction patterns (illustrated in Figure 10) with ferrite and martensite phase. The Rp, Rwp and χ2 profile residual factors are customarily used to characterize both the full pattern decomposition and Rietveld refinement quality. In all cases, the profile residual factors had smaller values for the refinements with the martensite phase.
Reviewer: May it is right what they see in Figure 8 "hard and brittle carbides" but they should validated by EDS otherwise is not valid info.
Response: We added EDS observation into manuscript to validate our statement. Moreover, carbides presences were confirmed by XRD technique.
Manuscript: We added the text. Figure 9 represents the chemical analysis of the martensite matrix as well as the embedded carbide particle. It can be clearly proved that the particle embedded in the matrix should be referred as carbides due to increased intensity of carbon as well as chromium comparing with the martensite matrix. Such finding also proves XRD patterns as those depicted in Figure 10.
Reviewer: How was measured this "The measured volume of retained austenite after HT (before hard milling) is 19%."? because I don not agree the OM measurements.
Response: It is indicated in chapter 2. It was measured by the use of XRD technique (Rietveld method). You can also see the austenite diffraction peaks in the Fig. 10. Please consider that due to the low thickness of the preferentially oriented near surface region (as the main source of massive MBN emission) the GID technique was applied. The effective penetration depth in such case is about 1 mm and therefore it is difficult to find correlation with OM.
Manuscript: Added Fig. 10. The Rietveld refinement was used for calculation of the phase composition as well as the mass percentage of retained austenite.
Reviewer: This part "Matrix precipitates, grain boundaries...already presented closure domains." is not supported by any evidence literature or authors measurements.
Response: We added citations in which theoretical models as well as real closure domains were observed especially nearby carbides.
Manuscript: Matrix precipitates, grain boundaries and free surfaces usually produce magnetic dipoles [29, 30]. Then, the secondary domain structure in the form of closure (reversed) domains can be found at the boundary of precipitates [31] (or other non-ferromagnetic particles) and the matrix, on the free surface [29, 30] and grain boundaries (to reduce or diminish the magnetostatic energy). Closure domains decrease in size with increasing magnetic field; however, some of them could retain the domain structure despite very high magnetic fields being employed [31]. As soon as the magnetic field is reversed, domain subsequent growth occurs in the opposite direction and BW motion is preferentially initiated on the already presented closure domains.
Reviewer: How was calibrated "Barkhausen noise " for example sin Figure 10 ?
Response: MicroScan represents the well-known and professional system (StressTech, Finland) for Barkhausen noise measurement adapted for hundreds of the real industrial application over the world. The device used in our study is calibrated each year by a professional person approved for such task (Mr. Malec – PCS Prague). Moreover, the sensor MBN response was checked by the use of the special sintered disk developed for such purpose (in the Stresstech – Finland). This disk gives the constant MBN emission. The magnetic field strength was measured by the use of Gauss meter.
As you indicate in your review the sine profile of magnetizing current is gently distorted. We know about that and we reported about it our study
D. Blažek, M. Neslušan, M. Mičica, J. Pištora, Extraction of Barkhausen noise from the measured raw signal in the high-frequency regimes, Measurement 94 (2016) 456-463, doi: 10.1016/j.measurement.2016.08.022.
Magnetizing current profile is distorted due to magnetic oversaturation of the exciting core. This distortion occurs as soon as the magnetizing current exceeds 110 mA. Such distortion has negligible influence on MBN emission since magnetizing current distortion occurs first of all during saturation phase of magnetizing cycle where BN emission is only a minor whereas BN emission is initiated near the coercive field when the magnetizing current is close to zero.
Manuscript: We prefer no change.
Reviewer: Again "Figure 8 and Figure 11 also show that VB and the corresponding alteration in the microstructure" they stated about microstructure but not clear evidence.
Response: We agree. It was not really clear. For this reason we updated the Fig. 8 and gently modify the text.
Manuscript: Figure 8, Figure 11b and Figure 13 also show that VB and the corresponding alteration in the near surface region (considering the preferential orientation as well as the mass percentage of retained austenite) take a significant role. One might expect that MBN in the tangential direction would increase along with increasing VB since the preferential orientation of the martensite matrix penetrates deeper. However, this aspect is strongly compensated by the increasing retained austenite volume (see Figure 11b). On one hand, increasing retained austenite volume decreases the ferromagnetic martensite volume as the phase contributing to the MBN signal. On the other hand, retained austenite as a non-ferromagnetic phase that strongly pins BW motion in the ferromagnetic phase. Comparing Figure 11b and Figure 13, it can be seen that MBN in the tangential direction is inversely proportional to the volume of retained austenite.
Reviewer: The conclusion doesn't express clear feedback for this research.
Response: We agree and therefore we added edited more extensive conclusions.
Manuscript: The acceptable surface state expressed in MBN threshold depends on the functionality of components being hard milled. The possible concept in which a hard-milled surface would be monitored is driven by the relationship between MBN and thickness of the near-surface region altered by the cutting process. As opposed to grinding, the high MBN values and degree of magnetic anisotropy should be linked with the low thickness of altered layers, whereas a progressive decrease of MBN should be linked with the thicker region of the near-surface containing increasing volume of retained austenite and undergoing severe plastic deformation at elevated temperatures. Due to the preferential orientation of the near surface region (tangential direction) and the corresponding BWs alignment MBN in this layer occurs in the form of a single massive MBN event. The remarkably lower MBN for the axial direction is associated with the nucleation and subsequent growth of 90° BWs as well as the process of 180° BWs rotation in the WL. The hysteresis loops as well as the polar graphs prove that a small change altering the magnetic field initiates a very early abrupt change of magnetisation as a result of irreversible domain wall jump in the tangential direction whereas the coherent BWs rotation combined with two-step switching can be found for the axial direction.
Notification:
It should be considered that the main focus of this study is in MBN emission and detail analysis of domain walls motion. We were asked by editor to contribute to the special issue of journal Materials focused in magneto-elastic behavior of materials. For this reason we did not provide too detailed view into microstructure only the aspects significantly related to the MBN emission. From our point of view the most valuable contribution of this study should be viewed in deep analysis of specific mechanism of domain walls motion, domain walls rotation and unusually high MBN emission in quenched surface.

Reviewer 2 Report
The article describes the characterisation of a hard-milling process by means of X-ray diffraction, Barkhausen noise measurements and observations based on the KERR-effect. The motivation of the study is to characterise the hard-milling process and the material parameter relating thereto by the means of Barkhausen noise measurements and to separate the influence by residual stress-, remaining austenite measurements and description of the magnetisation behaviour.
The study has a good scientific approach, with comprehensive characterisation methods. The experimental procedure and the results are presented in a comprehensible manner. To improve the article the conclusion may be more extensive.
Line 95: “2” should be written as “two”
Line 105: RS is not introduced.
Figure 7: Why is the colour of the mounting resin different?
Figure 8: Different magnifications make it difficult to compare the micrographs
Contaminations can be seen in both images
Recommend further publications for integration in “References”:
Baak, N.; Schaldach, F.; Nickel, J.; Biermann, D.; Walther, F. Barkhausen Noise Assessment of the Surface Conditions Due to Deep Hole Drilling and Their Influence on the Fatigue Behaviour of AISI 4140. Metals 2018, 8, 720.
Santa-aho, S.; Vippola, M.; Sorsa, A.; Leiviskä, K.; Lindgren, M.; Lepistö, T.:
Utilization of Barkhausen noise magnetizing sweeps for case-depth detection from hardened steel. NDT&E Int. 2012, 52, 95–102.
Author Response
Response to the reviewers
All changes made in the manuscript (additional texts and corrections) are highlighted yellow color (valid for the manuscript as well as this document).
Reviewer n. 2:
The article describes the characterization of a hard-milling process by means of X-ray diffraction, Barkhausen noise measurements and observations based on the KERR-effect. The motivation of the study is to characterize the hard-milling process and the material parameter relating thereto by the means of Barkhausen noise measurements and to separate the influence by residual stress-, remaining austenite measurements and description of the magnetization behavior.
The study has a good scientific approach, with comprehensive characterization methods. The experimental procedure and the results are presented in a comprehensible manner.
Reviewer: To improve the article the conclusion may be more extensive.
Response: We agree and therefore we added edited more extensive conclusions.
Manuscript: The acceptable surface state expressed in MBN threshold depends on the functionality of components being hard milled. The possible concept in which a hard-milled surface would be monitored is driven by the relationship between MBN and thickness of the near-surface region altered by the cutting process. As opposed to grinding, the high MBN values and degree of magnetic anisotropy should be linked with the low thickness of altered layers, whereas a progressive decrease of MBN should be linked with the thicker region of the near-surface containing increasing volume of retained austenite and undergoing severe plastic deformation at elevated temperatures. Due to the preferential orientation of the near surface region (tangential direction) and the corresponding BWs alignment MBN in this layer occurs in the form of a single massive MBN event. The remarkably lower MBN for the axial direction is associated with the nucleation and subsequent growth of 90° BWs as well as the process of 180° BWs rotation in the WL. The hysteresis loops as well as the polar graphs prove that a small change altering the magnetic field initiates a very early abrupt change of magnetisation as a result of irreversible domain wall jump in the tangential direction whereas the coherent BWs rotation combined with two-step switching can be found for the axial direction.
Reviewer: Line 95: “2” should be written as “two”
Response: corrected
Manuscript: MBN was measured in two perpendicular directions …..
Reviewer: Line 105: RS is not introduced.
Response: corrected
Manuscript: To determine residual stresses (RS), the Winholtz and Cohen ...
Reviewer: Figure 7: Why is the color of the mounting resin different?
Response: We updated images in Fig. 7. However it should be considered that hard machining process produces quite low WL (or HAZ) for low VB (usually difficult to observe in OM – thin and discontinuous). Therefore for better illustration we decided to provide OM mainly for samples milled by the inserts of high VB = 0.6 and 0.8 mm (thicker WL and HAZ) and details in the form of SEM images in Fig. 8. We tried to find the position on the sample surface in which the thickness of WL attains the maximum for better illustration.
MBN emission after hard milling is mainly a function of the thin near surface layer severely strained in the direction of cutting speed (tangential direction). For this reason, we provide details in the form of SEM images about extent of remarkable preferential orientation of the machined surface. Finally it can be found that the thickness of WLs on OM is more than the thickness of preferentially oriented surface. Preferential orientation of the matrix is due to elevated temperatures and superimposing plastic deformation whereas WL is due to predominating temperature cycle as a result temperatures exceeding austenitizing temperature followed by rapid cooling. Influence of severe plastic deformation in machining is limited whereas elevated temperatures penetrates much deeper due to good thermal conductivity of the bearing steel.
Manuscript: Please check the appearance of optical images in Fig. 7 and Fig. 8.
Reviewer: Figure 8: Different magnifications make it difficult to compare the micrographs
Response: Yes, we agree – the magnifications are really different. Therefore we updated images in Fig. 8.
Manuscript: Please check the appearance of images in Fig. 8.
Reviewer: Contaminations can be seen in both images
Response: Yes, we agree – corrected. We updated images in Fig. 8.
Manuscript: Please check the appearance of images in Fig. 8.
Reviewer: Recommend further publications for integration in “References”:
Baak, N.; Schaldach, F.; Nickel, J.; Biermann, D.; Walther, F. Barkhausen Noise Assessment of the Surface Conditions Due to Deep Hole Drilling and Their Influence on the Fatigue Behaviour of AISI 4140. Met. 2018, 8, 720.
Santa-aho, S.; Vippola, M.; Sorsa, A.; Leiviskä, K.; Lindgren, M.; Lepistö, T.:
Utilization of Barkhausen noise magnetizing sweeps for case-depth detection from hardened steel. NDT&E Int. 2012, 52, 95–102.
Response: we integrated the aforementioned publication in the manuscript main body as well as “references”
Manuscript: please check list of references (references 3 and 4) as well as manuscript main body.
… MBN is investigated as a function stress state [1 ÷ 4], dislocation…

Round 2
Reviewer 1 Report
The authors considered the revision and honestly argumented their work, therefore I suggest accepting this paper